# Universal Modal Embedding of Dynamics in Videos and Its Applications

## Abstract

Extracting underlying dynamics of objects in image sequences is one of the challenging problems in computer vision. On the other hand, dynamic mode decomposition (DMD) has recently attracted attention as a way of obtaining modal representations of nonlinear dynamics from (general multivariate time-series) data without explicit prior knowledge about the dynamics. In this paper, we propose a convolutional autoencoder based DMD (CAE-DMD) that is an extended DMD (EDMD) approach, to extract underlying dynamics in videos. To this end, we develop a modified CAE model by incorporating DMD on the encoder, which gives a more meaningful compressed representation of input image sequences. On the reconstruction side, a decoder is used to minimize the reconstruction error after applying the DMD, which in result gives an accurate reconstruction of inputs. We empirically investigated the performance of CAE-DMD in two applications: background/foreground extraction and video classification, on publicly available datasets.

## 1 Introduction

Extracting underlying dynamics of objects in video frames is one of the challenging problems in video processing Erichson et al. (2016). Meanwhile, dynamic mode decomposition (DMD) Rowley et al. (2009); Schmid (2010) has recently attracted attention as a way of obtaining modal representations of nonlinear dynamics from (general multivariate time-series) data without explicit prior knowledge about the dynamics. It is closely related to the spectral analysis of the Koopman operator and has been successfully applied for the extraction of spatiotemporal patterns which is crucial in many applications of engineering fields, such as fluid dynamics, epidemiology, neuroscience, controlled systems, analysis of power systems, oceanography, molecular kinetics and many others. We consider two major applications which are based on the underlying dynamics of videos, one is the background/foreground extraction and the other is video classification. Since DMD can extract low rank spatio-temporal features of complex dynamical systems and gives information of growth rates and frequencies of the dynamics, which allows us to interpret spatial structural and temporal information in the data. Therefore, it is possible to perform analysis in a dimensionality reduction data by considering dynamics embedding with the modes obtained by DMD. However, standard or exact DMD method is unable extract the complex non-linear dynamics in time series data, since in the standard approach vertically aligned spatio-temporal image sequences are processed all at once with any further processing, so it becomes difficult to extract the required dynamic information from the data.

In this paper, we propose a universal modal embedding of dynamics in videos by CAE-DMD method; a variant of extended dynamic mode decomposition (EDMD) to obtain a compact modal representations of non-linear dynamics from a general multivariate time-series data. In CAE-DMD, at first latent vectors of complex video sequences are obtained by training a CAE network and then DMD is applied on these latent vectors to obtain modal representation of input image sequences, those contain the spatial information of respective image sequences. This method exploits the spatial and temporal information in a low-rank modal with latent vector representations that gives more meaningful representation of input image sequences. We investigate the empirical performance of CAE-DMD on publicly available datasets and results show that our method achieves competitive performance and can cope with complex dynamics in videos or any time-series data.

---

**Algorithm 1** Dynamic Mode Decomposition Schmid (2010)

---

**Require:** $V_1$ and $V_2$ input sequences.
**Ensure:** Dynamic modes $\Phi$ and eigenvalues $\Delta$.
 1: $U_r, S_r, Q_r \leftarrow$ compact SVD of $V_1$.
 2: $\tilde{A} \leftarrow U_r^* V_2 Q_r S_r^{-1}$.
 3: $\tilde{W}, \Delta \leftarrow$ eigenvectors and eigenvalues of $\tilde{A}$;
 4: $\Phi \leftarrow V_2 Q_r S_r^{-1} \tilde{W}_r$
 5: **return:** $\Phi, \Delta$;

---

The remainder of this paper is organized as follows: First, we give a brief review on Koopman spectral analysis and Extended DMD in Section 2. In Section 3, we describe our proposed model, and video processing with CAE-DMD is presented in Section 4. Experiments and results are discussed in in Section 5 along with performance evaluations on the basis of benchmark datasets. Finally, Section 6 summarizes and concludes the study.

## 2   BACKGROUND: EXTENDED DMD

Consider a (possibly nonlinear) dynamical system:

$$v_{t+1} = f(v_t), \quad v \in \mathcal{M},$$

where $f\colon \mathcal{M} \to \mathcal{M}$, $\mathcal{M}$ is the state space, and $t$ is the time index. In this system, the Koopman operator $\mathcal{K}$ for $\forall v \in \mathcal{M}$ can be defined as follows:

$$\mathcal{K}g(v) = g(f(v)),$$

where $g\colon \mathcal{M} \to \mathbb{C}$ ($\in \mathcal{F}$) denotes an observable that is an element in some function space $\mathcal{F}$ Koopman (1931). By definition, $\mathcal{K}$ is a linear operator in $\mathcal{F}$. Here, we assume that there exists a subspace of $\mathcal{F}$ invariant to $\mathcal{K}$, which can be denoted by $\mathcal{G} \subset \mathcal{F}$. Additionally, we assume that $\mathcal{G}$ is finite-dimensional and that a set of observables $\{g_1, \ldots, g_n\}$ that spans $\mathcal{G}$ is available. If we let $g = [g_1, \ldots, g_n]^\top \colon \mathcal{M} \to \mathbb{C}^n$, the one-step evolution of $g$ for $\forall v \in \mathcal{M}$ can be expressed as follows:

$$Kg(v) = g(f(v)),$$

where the finite dimensional $K$ is the restriction of $\mathcal{K}$ to $\mathcal{G}$. Then, if all eigenvalues of $K$ are distinct, any value of $g$ can be expressed as follows:

$$g(v) = \sum_{i=1}^n \varphi(v)\xi_i,$$

with some coefficients $\xi_i$, where $\varphi\colon \mathcal{M} \to \mathbb{C}^n$ is an eigenfunction of $K$ with the corresponding eigenvalue $\lambda \in \mathbb{C}$, i.e., $K\varphi(v) = \lambda\varphi(v)$. Thus, we obtain

$$g(v_t) = \sum_{i=1}^n \lambda_i^t c_i, \quad c_i = \varphi_i(v_0)\xi_i.$$

As a result, $g$ is decomposed into modes $\{c_i\}$, and the modulus and argument of $\lambda_i$ express the decay rate and frequency of $c_i$, respectively.

Next, we summarize EDMD as a method to approximates the Koopman operator Williams et al. (2015). It requires a pair of time series data as

$$V_1 = [v_1, v_2, v_3, \ldots, v_{T-1}], V_2 = [v_2, v_3, v_4, \ldots, v_T],$$

where $v_t \in \mathcal{M}$ and $v_{t+1} = f(v_t)$. And then, suppose we are given a set of independent basis functions $\mathcal{B} = \{b_1, b_2, \ldots, b_n\}$, where $b_i \in \mathcal{K}$. A vector valued function $b\colon \mathcal{M} \to \mathbb{C}^{1 \times n}$, where $b(v) = [b_1(v), b_2(v), \ldots, b_n(v)]$. Next, we generate $K \in \mathbb{R}^{n \times n}$, a finite dimensional approximation of $\mathcal{K}$ and a function $b_d \in \mathcal{F}_b \subset \mathcal{F}$, written as

$$b_d = \sum_{i=1}^n w_i b_i = bw,$$

where $w$ contain the weights of $n$ elements of $\mathcal{B}$. Since $\mathcal{F}_b$ is not an invariant subspace of $\mathcal{K}$, Therefore

$$\mathcal{K}b_d = b(Kw) + r.$$

The above equation includes a residual term $r \in \mathcal{F}$. Therefore, to find $\boldsymbol{K}$, we minimize the following optimization

$$J = \frac{1}{2} \sum_{i=1}^{T-1} |(\boldsymbol{b}(\boldsymbol{v}_{i+1}) - \boldsymbol{b}(\boldsymbol{v}_i)\boldsymbol{K})\boldsymbol{a}|,$$

which is a least square problem whose unique solution can be found as,

$$\boldsymbol{K} = \boldsymbol{G}^{\dagger}\boldsymbol{A},$$

where $\dagger$ denotes the pseudoinverse, while $\boldsymbol{G}$ and $\boldsymbol{A}$ can be obtained by solving the following equations

$$\boldsymbol{G} = \frac{1}{T-1} \sum_{i=1}^{T-1} \boldsymbol{b}(\boldsymbol{v}_i) * \boldsymbol{b}(\boldsymbol{v}_i),$$

$$\boldsymbol{A} = \frac{1}{T-1} \sum_{i=1}^{T-1} \boldsymbol{b}(\boldsymbol{v}_i) * \boldsymbol{b}(\boldsymbol{v}_{i+1}),$$

where $\boldsymbol{K}, \boldsymbol{G}, \boldsymbol{A} \in \mathbb{C}^{n \times n}$. Thus, $\boldsymbol{K}$ is a finite dimensional approximation of $\mathcal{K}$ and EDMD approximation of an eigenfunction of $\mathcal{K}$ can be written as

$$\varphi_i = \boldsymbol{b}\xi_i.$$

In the above equation $\xi_i$ is the $i$-th eigenvector of $\boldsymbol{K}$. Note that here we briefly summarized EDMD, a more detailed explanation can be found in Williams et al. (2015).

## 3 PROPOSED MODEL: CAE-DMD

In the propose model of CAE-DMD, we first train the covolutional auto-encoder network on input image sequences to obtain the compact representation of these sequences in the form of latent vectors/encoded sequences. These encoded sequences of input videos capture the complex underlying dynamics after training the CAE network. In the next step, we apply DMD on these compact representations to obtain modal representations, those contain the spatial information of image sequences. Most of the spatial information of video sequences can be found in few DMD modes. Therefore, a few set of modes can capture dynamics of even longer video sequences, that is one of the main advantages of dimensionality reduction temporally using DMD. Also, the spatial dimension of input sequences can be reduced by choosing the length of latent vectors. Therefore, we exploit the dimensionality reduction spatially and temporally for modal embedding of dynamics in time series sequences. Also, extraction of non-linear information in the data is achieved by representing the input image sequences in the form of encoded vectors using deep convolutional autoencoder. The architecture of CAE-DMD for foreground and background extraction is illustrated in Figure 1. The details of the main blocks of CAE-DMD model are described as follows:

**Encoder:** In the first step, input video sequences $\{\boldsymbol{v}_1, \boldsymbol{v}_2, \ldots, \boldsymbol{v}_{T-1}\}$ are trained on convolutional autoencoder network. For this training we used sixteen layers deep CAE network. The filter size of each convolutional layer was fixed to $5 \times 5$ with increasing number of total filters at each layer. The Encoder takes flattened video sequences as input and generates the feature map in the next layer as

$$\mathbf{z}_i = f(\mathbf{p}_i \times \mathbf{V}_1 + \mathbf{b}_i), \tag{1}$$

where $\mathbf{V}_1$ contains the input images, $\mathbf{p}_i$ and $\mathbf{b}_i$ are the corresponding filter and bias, and $f$ is the activation function. In this model, we used Rectified Linear Unit (ReLU) as activation function after each convolutional layer Zeiler et al. (2013), which is further followed by a max pooling layer. This helps reducing the computational cost in the upper layers. Then, the latent spaces are obtained by a fully connected layer at the output of encoder as

$$\mathbf{H} = \mathbf{Z} \times \mathbf{W} + \mathbf{C}. \tag{2}$$

In equation 2, $\mathbf{H}$ denotes the hidden layer which contains the compressed sequences of input images, $\mathbf{Z}$ represents the feature maps, whereas $\mathbf{W}$ and $\mathbf{C}$ are the feature weights and bias of the fully connected layers, respectively. Note that in this model we choose the length of a latent vector to half

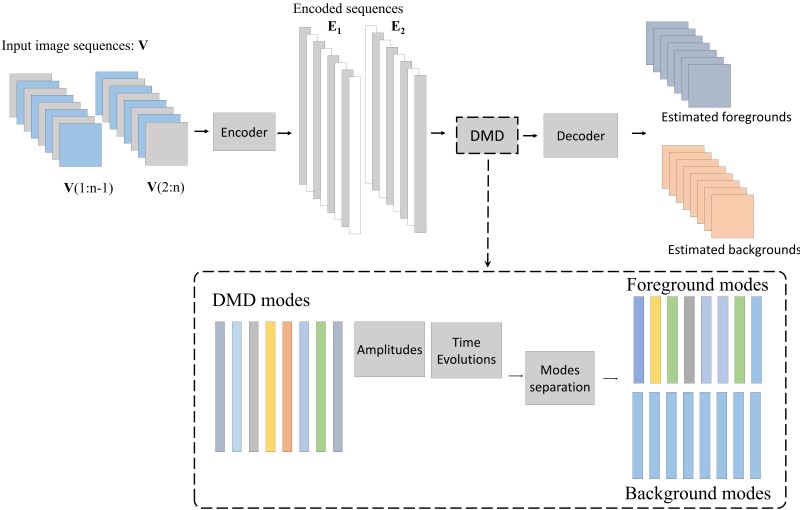

Figure 1: Convolutional autoencoder block diagram for foreground/background extraction.

the size of an input image, which gives promising results to extract dynamics from image sequences and for classification of videos.

**Decoder:** The proposed model of decoder follows a symmetric pattern of layers as in the encoder. To reconstruct the input images at the decoder side, the latent sequences obtained from the encoder are reshaped to a reconstructed version of feature maps $\mathbf{Z}'$ via fullyconnected layer as

$$\mathbf{Z}' = \mathbf{H} \times \mathbf{W}' + \mathbf{C}', \tag{3}$$

where $\mathbf{W}'$ and $\mathbf{C}'$ are the weights and bias of fullyconnected layer at the decoder side. Then, up-sampling is applied after each convolutional layer and the final approximated image $\tilde{V}_1$ is reconstructed at the output of the last layer of Decoder by reshape function. Note that too much compression in CAE will result in loss of information. Typically a simple loss function; Mean Square Error (MSE) is calculated for convergence of the CAE model between the network output $\tilde{V}_1$ i.e., output of decoder and the corresponding vectorized original video sequences $\boldsymbol{V}_1$. However, in the proposed method DMD loss term is introduced and a modified objective function is implemented for end-to-end learning.

$$MSE = \frac{1}{b_s} \sum_{i=1}^{T/b_s} \|\boldsymbol{v}_{bs}i - \tilde{\boldsymbol{v}_{bs}i}\|^2 + \lambda_d \|\boldsymbol{H}_2 - \boldsymbol{U}\tilde{\boldsymbol{A}}\boldsymbol{U}'\boldsymbol{H}_1\|^2, \tag{4}$$

where $T$ is the total number of video sequences and $b_s$ is the batch size for training. The incorporated DMD loss term in the objective function is used to minimize the DMD reconstruction error that is controlled by the parameter $\lambda_d$. $\boldsymbol{H}_1$ and $\boldsymbol{H}_2$ are the encoded sequences and shifted encoded sequences in time step, respectively. $\boldsymbol{U}$, $\boldsymbol{U}'$ are orthogonal matrices and $\tilde{\boldsymbol{A}}$ is a low-dimensional linear operator obtained by applying DMD on latent vectors (see Alg. 1).

**DMD over Encoded Sequences:** In the next step, original image sequences $\{\boldsymbol{v}_1, \boldsymbol{v}_2, \ldots, \boldsymbol{v}_{T-1}\}$ and $\{\boldsymbol{v}_2, \boldsymbol{v}_3, \ldots, \boldsymbol{v}_T\}$ are fed to the trained CAE network to obtain the latent vectors of input image sequences. The dimension of these latent vectors is spatially compressed as compared to input sequences, DMD is applied on these vectors to obtain dynamic modes, those contain the complex dynamic with low and high frequency information of the data with their corresponding eigenvalues. These filtered latent vectors are further fed to the decoder for final approximation of video sequences.

## 4 VIDEO PROCESSING WITH CAE-DMD

For a given video frames $\mathbf{V} \in \mathbb{R}^{n_1 \times n_2 \times T}$, each frame $\{\boldsymbol{v}_1, \boldsymbol{v}_2, \ldots, \boldsymbol{v}_T\}$ is flattened as column vector and ordered in time. This arrangement of video frames gives a spatiotemporal grid, that is arranged

in two overlapping sets of data called left and right image sequences defined as:

$$V_1 = \left[v_1, v_2, \ldots, v_{T-1}\right], V_2 = \left[v_2, v_3, \ldots, v_T\right]. \tag{5}$$

To obtain the DMD of a given video, it is assumed that its each frame $v_{t+1}$ at time interval (t+1) is connected to the previous frame $v_t$ by linear mapping $\tilde{A}$ as $V_2 = \tilde{A}V_1$, the estimate of $\tilde{A}$ and its eigen decomposition is obtained by Alg. 1.

A set of dynamic modes $\boldsymbol{\Phi} := \{\phi_1, \ldots, \phi_r\}$ and the corresponding eigenvalues $\boldsymbol{\Delta} := \{\Lambda_1, \ldots, \Lambda_r\}$ obtained are used to reconstruct these image sequences $\tilde{V}_1$ and $\tilde{V}_2$ which are the approximation of original video sequences. Here, $r$ is the number of adopted eigenvectors. These modes represent the slowly varying or rapidly moving objects at time points $t \in \{0, 1, 2 \ldots, T-1\}$ in the video frames with associated continuous-time frequencies and can be expressed as follows:

$$\boldsymbol{\omega}_j = \frac{\log(\Lambda_j)}{\Delta t}. \tag{6}$$

Further, the approximated video frames for low- and high-frequency modes at any time point can be reconstructed as

$$\tilde{V}(t) \approx \sum_{j=1}^{r} \phi_j \exp(\boldsymbol{\omega}_j t)\boldsymbol{\alpha}_j = \boldsymbol{\Phi}\exp(\boldsymbol{\Omega}t)\boldsymbol{\alpha}, \tag{7}$$

where $\phi_j$ is a column vector of the $j$-th dynamic mode that contains the spatial structure information and $\boldsymbol{\alpha}_j$ is the initial amplitude of the corresponding DMD mode. The vector of the initial amplitudes $\boldsymbol{\alpha}$ can be obtained by taking the initial video frame at time $t = 0$, which reduces equation 7 to $v_1 = \boldsymbol{\Phi}\boldsymbol{\alpha}$. Note that the matrix of eigenvectors is not square; thus, the initial amplitudes can be observed using the following pseudoinverse process:

$$\boldsymbol{\alpha} = \boldsymbol{\Phi}^\dagger v_1. \tag{8}$$

Table 1: F-measures by CAE-DMD and the existing algorithms for foreground extraction on Wallflower dataset.

| Dataset | DECOLOR | 3TD | DP-GMM | LSD | TVRPCA | SRPCA | LR-FSO | MODSM | GFL | RFSA | GRASTA | MSCL-FL | CAE-DMD |
|---|---|---|---|---|---|---|---|---|---|---|---|---|---|
| Wallflower | 0.59 | 0.75 | 0.78 | 0.75 | 0.61 | 0.85 | 0.74 | 0.73 | 0.84 | 0.54 | 0.33 | 0.92 | 0.87 |

## 4.1 FOREGROUND INFORMATION EXTRACTION

Foreground segmentation and background estimation has been originated in many computer vision applications such as moving object detection Zhou et al. (2013), video surveillance Bouwmans & Zahzah (2014), motion detection Gao et al. (2014) and many others. A number of methods have been proposed in literature to achieve this task. Ortego et al. (2016) proposed multipath reconstruction for background estimation. Liu et al. (2015) estimated the background and foreground based on low-rank and structured sparse matrix, respectively. The performance of these methods degrades if the background is visible for a short period or in case of dynamic background. To overcome these limitations Javed et al. (2017) proposed a method to incorporate the spatial and temporal sparse subspace clustering into the Robust Principal Component Analysis (RPCA) Candès et al. (2011). However in our proposed CAE-DMD method, we split the foreground and background based on the DMD modes and their corresponding eigenvalues (frequencies). Thus one can extract not only the background or foreground information but also other dynamics in the data, based on the frequency information obtained after applying DMD on latent vectors. Figure 3 shows the case of splitting foreground and background in the video.

The key principle to separate the video frames into foregrounds and the background is by thresholding of low frequency modes based on the corresponding eigenvalues. Generally, the portion that represents the background in videos is constant among the frames and satisfies $|\boldsymbol{\omega}_p| \approx 0$, where $p \in \{1, 2, \ldots, r\}$. Typically, a single mode represents the background, which is located near the origin in complex space, whereas $|\boldsymbol{\omega}_j|, \forall j \neq p$ are the eigenvalues that represent the foreground

structures bounding away from the origin in complex plane. Therefore, the reconstructed video frames can be separated into the background and foreground structures as follows:

$$\mathbf{V}_{DMD} = \underbrace{\boldsymbol{\phi}_p \exp(\boldsymbol{\omega}_p t)\boldsymbol{\alpha}_p}_{Background} + \underbrace{\sum_{j \neq p} \boldsymbol{\phi}_j \exp(\boldsymbol{\omega}_j t)\boldsymbol{\alpha}_j}_{Foreground}, \tag{9}$$

where $t = \{0, \ldots, T-1\}$ is the time indices up to $(T-1)$ frames. Note that the initial amplitude $\boldsymbol{\alpha}_p = \boldsymbol{\phi}_p^\dagger \{\boldsymbol{v}_1\}$ of the stationary background is constant for all the future time points, whereas $\boldsymbol{\alpha}_j = \boldsymbol{\phi}_j^\dagger \{\boldsymbol{v}_1\}, \forall j \neq p$ are the initial amplitudes of varying foreground structures.

After filtering the DMD modes based on the eigenvalues, encoded sequences are reconstructed and fed to the decoder to approximate the background and foreground structures in the video sequences. Figure 3 shows a set five frames of Wallflower dataset (Bootstrap video sequences) of moving people. Figure 3 second row shows the obtained eigenvalues each of which corresponds to either foreground or background modes.

### 4.2 CLASSIFICATION OF VIDEOS

To extract the dynamics in videos several methods have been proposed for different applications such as dynamic scene classification Derpanis et al. (2012), activity and action recognition Donahue et al. (2015) in videos. Video classification using Convolutional Neural Networks (CNNs) have been considered as an effective class of models, where the networks not only learn the appearance information but also able to learn temporal evolution Karpathy et al. (2014); Du et al. (2015). In these methods the temporal encoding is learned at a single level that is not sufficient to interpret and understand the complex dynamics in video sequences. A more well suited method is proposed by Cherian et al. (2017), they proposed a rank pooling method that takes input, features from the intermediate layer of (CNN) trained on tiny subspaces. They use this low-rank approximation of features while maintaining temporal order for compact representation of video sequences and then use this for classification. A recently introduced discriminative hierarchical rank pooling method Fernando et al. (2016) cope with the limitations in previous methods and proposes a novel method to encode video sequences at multiple levels to capture the non-linear and complex dynamics. Some of the famous traditional methods for one activity classification per video are: bag of features (BoF), kernel dynamic system (KDS) Laptev et al. (2008), binary dynamic system (BDS) Niebles et al. (2010), DTM Blei & Lafferty (2006), Attributes based LI & Vasconcelos (2012) and TOT those are explained by LI & Vasconcelos (2012) .

In our proposed method, we classify the complex dynamics in videos through dynamic modes, obtained by applying the DMD on latent spaces of input video sequences. In the next step of video classification, orthogonal vectors of these DMD modes are calculated by applying singular value decomposition (SVD) and then a projection kernel Hamm & Lee (2008) is computed on these orthogonal vectors. Finally, a supervised classifier is trained over this kernel with respective labels of videos. The method to classify videos along with the projection kernel method is summarized in Alg. 2.

## 5 EXPERIMENTAL RESULTS

### 5.1 EXTENDED (CAE-DMD) AND STANDARD DMD

To demonstrate the effectiveness of the proposed method, another experiment is performed on a video of SBMnet[1] dataset, where people are strolling in a terrace with no original background provided, so the extraction of dynamics becomes more challenging. To visualize and verify our method we extracted foreground information from a set of 150 consecutive image sequences and then applied both methods i.e., standard and CAE-DMD. Results of standard and CAE-DMD are shown in Figure 2 (a) and Figure 2 (b), respectively. It can be visualized in Figure 2 second row, that standard DMD method is unable to reconstruct original image sequences because of the presence of non-linear complex dynamics in video frames. On the other side, CAE-DMD can reconstruct

---

[1]http://scenebackgroundmodeling.net/

---

**Algorithm 2** CAE-DMD to classify videos.

---

**Require:** $D$ and $L$, DMD modes and videos labels, respectively.
**Ensure:** $\tilde{L}$ predicted label.
 1: $U \leftarrow$ compact SVD of $D$.
 2: **for** $i = 1$ to $ns$ **do**                                             ▷ $ns$ : total number of videos to train.
 3:     **if** $i \geq 2$ **then**
 4:         **for** $j = 1$ to $i - 1$ **do**
 5:             $K(i,j) = \text{real (trace } (U_j * U_j^*)(U_i * U_i^*))$
 6:             $K(j,i) = K(i,j)$                                   ▷ $K$: kernel
 7:         **end for**
 8:     **end if**
 9: **end for**
10: Train KNN classifier over $K$ and $L$.                        ▷ KNN: k-nearest neighbors
11: $\tilde{L} \leftarrow$ predict with test data.

---

original image sequences more accurately, which shows that by applying DMD on latent vectors can effectively minimize the reconstruction error; that shows the method is capable of handling the non-linearity in data. Further, the dynamic and static information in the video can easily be separated with CAE-DMD shown in Figure 2 third-row. Figure 2 fourth-row shows the threshold image sequences after foreground extraction.

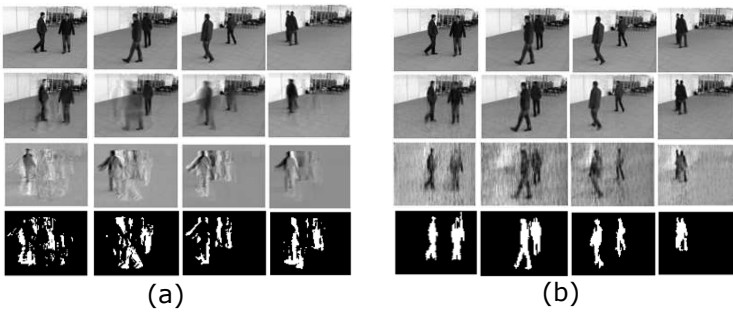

(a)                      (b)

Figure 2: Dynamics extraction in Standard and Extended (CAE) DMD; (a) Standard DMD. (b) CAE-DMD. First row: original Image sequences of walking people; Second row: Image reconstruction by standard and CAE-DMD; Third row: dynamics extraction; Last row: Threshold image sequences.

We also empirically investigated the performance of CAE-DMD based on comparisons with several existing algorithms in the scenarios of foreground extraction in Subsection 5.2, and video classification in Subsection 5.3.

**Experimental details:** We utilize convolutional autoencoder network to represent each video frame by a latent vector obtained by a fully connected layer, the length of each latent vector is set to half of the input image size, with a filter size of 5x5 and stride of one, unless otherwise specified. The network is implemented in python with Tensorflow package and parameters such as batch size of 16, learning rate $1e^{-4}$, number of epochs 500 1000, and optimization is done with adam optimizer. To avoid overfitting a dropout of 0.5 was used. Test, validation and train data is separated as 10%, 10% and 80%, respectively for foreground extraction tasks. Note that the optimal values of these parameters were manually tuned and set to those where the best performance was achieved. Total time to reconstruct a batch of images was 1-2.5s, after training of video frames.

## 5.2 FOREGROUND EXTRACTION

We applied the proposed method to extract foreground structures from the background using the *wallflower* dataset.[2] This dataset contains six video sequences (Moved Object, Camouflage, Foreground Aperture, Light Switch, Time of Day and Waving Trees) and a single hand segmented ground-truth image for each sequence. The results are compared with the state-of-the-art methods, such as DECOLOR Zhou et al. (2013), 3TD Oreifej et al. (2013), DP-GMMHaines & Xiang

---

[2]http://research.microsoft.com/en-us/um/people/jckrumm/WallFlower/TestImages.htm

(2014), LSD Liu et al. (2015), TVRPCA Cao et al. (2016), SRPCA Javed et al. (2018), LR-FSO Xue et al. (2013), MODSM, GFL Xin et al. (2015), RFSA Guo et al. (2014), GRASTA He et al. (2012) and MSCL-FL Javed et al. (2017). Figure 4 shows some examples of results by CAE-DMD for foreground extraction. The first column shows the original frames of video sequences. The estimated backgrounds by CAE-DMD is shown in the second column followed by difference images in the third column. The estimated foreground images by our method and the hand segmented ground-truth images of corresponding videos are shown in fourth and fifth columns, respectively. The difference image was calculated by considering the absolute difference between the original frames and their respective backgrounds. To further enhance the accuracy of the extracted foreground structures, morphological operations were applied to fill the empty holes and to connect the unconnected binary pixels followed by thresholding. The performance of CAE-DMD calculated by F-measure values are shown in Table 1, that shows the our method is competitive with other state-of-the-art proposed methods.

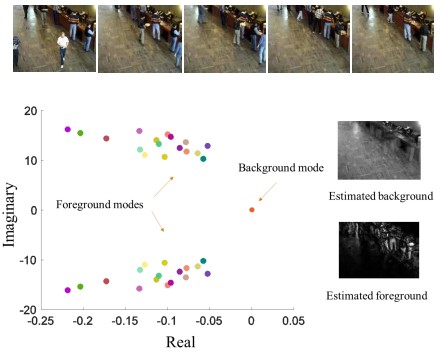

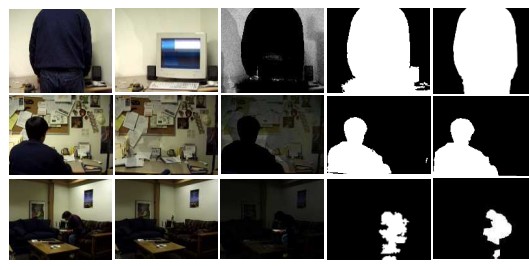

Figure 3: Splitting foreground and the background in Wallflower dataset (Bootstrap video sequences).

Figure 4: Wallflower dataset; first column frames of original videos; Estimated backgrounds are shown in the second column; Difference and threshold images are shown in column three and fourth respectively. Fifth column shows the enhanced images after applying morphological operations.

Table 2: Classification accuracy on Weizmann Activities.

| Dataset | BoF | ATTRIBUTES | DTM | TOT | KDS | BDS | CAE-DMD |
|---|---|---|---|---|---|---|---|
| **Weizmann Activities** | 57.8% | 72.6% | 84.6% | 88.2% | 90.2% | 94.8% | 91.1% |

## 5.3 VIDEO CLASSIFICATION

Our second experiment is the classification of videos based on the dynamics they contain. We used Weizmann dataset[3] to classify different actions in videos. We consider this dataset suitable for our experiments, since it contains one particular action per video with 10 classes of different actions *e.g.,* (Bend, Jack, Jump, Pjump, Run, Side, Skip, Walk, Wave1 and Wave2) performed by 9 different subjects. Next we classify these activities with a 9-fold leave-one-out-cross-validation (LOOCV), where for each trial the activities of one subject were used as test set and those of the remaining 8 as a training set. The classification accuracy of our method is presented in Table 2, that shows classification through modal representations using DMD on latent vectors can be used to classify videos containing different actions.

## 6 CONCLUSIONS

We proposed a universal modal embedding of dynamics by CAE-DMD that is an extended DMD method to extract complex underlying dynamics in general multivariate time series data. The experimental results show that our modal embedding method; a dimensionality reduction spatially and temporally can effectively extract the underlying dynamics in videos. To verify our method we consider two applications: foreground extraction and videos classification. Comparable performance achieved on these applications exhibit that this method can be applied on any multivariate-time series data to extract complex and non-linear dynamics.

---

[3]http://www.wisdom.weizmann.ac.il/ vision/SpaceTimeActions.html

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
