# OpenReview forum: "UNIVERSAL MODAL EMBEDDING OF DYNAMICS IN VIDEOS AND ITS APPLICATIONS"
_ICLR.cc/2020/Conference — Reject_

### Official Review · AnonReviewer1 · 2019-10-22
**Official Blind Review #1**

**Rating:** 3

**Review:**

The paper considers the problem of extracting the underlying dynamics of objects in video frames. The paper focuses on two major applications: background/foreground extraction and video classification. The paper proposes a method that first obtains latent vectors from a video sequence by training a neural network and then applies dynamic mode decomposition (by Schmidt, 2010).

The paper is not well written and even after reading it the second time I, unfortunately, have difficulties understanding the exact contributions and experiments.
Here are concrete examples that lead to this critique:
- Section 2: Letters are not defined (e.g., \mathcal F is not defined when first used), and sentences are not finished and/or do not make sense.
- Sec. 5.1: not clear whether the experiment is only done for one sequence or for many. If done on only one sequence, this is not sufficient to demonstrate that the method works well, if done on more than one then the results are not reported.
- Conclusion: States that ``this method can be applied to any multivariate-time series data to extract complex and non-linear dynamics''. That statement sounds overlay general given the experimental evaluation.

**Experience Assessment:**

I have read many papers in this area.

**Review Assessment: Checking Correctness Of Derivations And Theory:**

I assessed the sensibility of the derivations and theory.

**Review Assessment: Checking Correctness Of Experiments:**

I assessed the sensibility of the experiments.

**Review Assessment: Thoroughness In Paper Reading:**

I read the paper at least twice and used my best judgement in assessing the paper.

---

### Official Review · AnonReviewer2 · 2019-10-24
**Official Blind Review #2**

**Rating:** 3

**Review:**

I found the topic of this paper interesting and I believe I understand what the authors are trying to achieve but I'm afraid this was after several readings and I do think the paper could be presented differently that would make it more accessible. My suggestion would be to explain how the model will be applied first (identify the required properties) to motivate the need for the learned basis and then present the DMD as a method for providing a basis that meets the properties required. I acknowledge that different communities have different styles of presentation so apologies if this is just me.

First I would just like to check that I have understood correctly so please could the authors point out if I have missed something or misunderstood in the following?

Our goal is to establish a basis invariant to the video dynamics that can then be used, for example, to partition the video into parts with differing dynamics - e.g. foreground/background. To do this we need to identify such a basis from a specific video - we will use the collection of pairs of neighboring frames.

The Koopman operator acts on a differential system to identify a function space invariant to the dynamics. If we instantiate this with a finite number of dimensions we can essentially establish the invariance as an eigenvalue problem. From this and our pairs of successive videos we can establish a vector basis for the space and then project the video into this basis. The spectral properties of the coefficients of the projection will determine whether something is static (omega = 0) or transitory in the scene and these can be used to identify foreground and background.

Next there is the issue that this method operates in a linear domain with something like Gaussian noise which is not a good fit for image space videos so the authors propose to identify the dynamics in a linear latent space determined by an autoencoder to handle the non-linear mapping to image space.

I hope I have understood the main points?

If this is the case, I think that much more needs to be said about the second part, which is the essential novelty of the paper, with a discussion of the merits of different approaches and full details - at the moment there is just one small paragraph at the end of 4.2 which contains the majority of the contribution.

My main concern about the paper is that I find it very difficult to appreciate the efficacy of the method given the current presentation of the results. There are no error bars to ascertain significance for any of the results and the summarization of multiple experiments to a single percentage gives very little insight into where this method works and where it doesn't. There are a number of ways that a dynamic prior could be added to a latent space and it is unclear why we would expect this approach to be preferred given the evidence presented in the paper.

Other Notes:

I found that the notation is not always consistent and sometimes could be simplified - it is unclear whether some operators are convolutions or multiplications (vector or scalar). To me the asterisk does not represent straight forward multiplication but it might be being used for this?

Could Table 1 be placed in the experiments section rather than in the middle of page 5?

Do the authors mean half the number of pixels or half the edge size (e.g. a quarter of the area) in terms of the latent space?

Please can all equations be numbered so that they can be referred to - there are no equation numbers in all of section 2.

**Experience Assessment:**

I have read many papers in this area.

**Review Assessment: Checking Correctness Of Derivations And Theory:**

I assessed the sensibility of the derivations and theory.

**Review Assessment: Checking Correctness Of Experiments:**

I assessed the sensibility of the experiments.

**Review Assessment: Thoroughness In Paper Reading:**

I read the paper thoroughly.

---

### Official Review · AnonReviewer3 · 2019-10-29
**Official Blind Review #3**

**Rating:** 6

**Review:**

This paper presents an application of convolutional autoencoder networks and a nonlinear dynamic systems analysis method known as extended dynamic mode analysis (EDMD) to a data-driven analysis of multivariate time series.
The DMD method appears to be well-known in the physics community but is outside my area of expertise and unfortunately I have limited time to make a quick study of it.  However, from what I gather, it involves empirical approximation of a nonlinear dynamical system as a high-dimensional linear dynamical system, which in turn enables analysis in terms of eigendecomposition of the resulting linear operator, revealing basic modes of the dynamics.   In the proposed method, DMD is used in the latent representations of a convolutional autoencoder for image sequences.   The DMD objective is incorporated into the autoencoder training loss to minimize its reconstruction error.    The DMD is also used, by conditioning on the eigenvalues, to split the reconstruction into high frequency (quickly varying) foreground modes and low-frequency (slowly varying) background modes.   Although end-to-end training is mentioned, it is not made clear how the derivatives of the DMD decomposition are implemented, especially considering that the DMD involves an SVD, which can have unstable/ singular derivatives when two or more singular values are close to the same / exactly the same.   The resulting methods are applied to a foreground extraction and a classification tasks, and compared with numerous baselines.  It is not clear to me what the state of the art is on these tasks, but the proposed methods compare favorably to reported baselines, and the images of results look convincing.   However the experimental results seem a little thin and I would expect a more thorough study.  Overall the method looks very interesting.

Some complaints:
 -  the tables are a bit sloppy and should be formatted to fit in the document with normal sized fonts,
 - the images are too small to see well.


**Experience Assessment:**

I do not know much about this area.

**Review Assessment: Checking Correctness Of Derivations And Theory:**

I assessed the sensibility of the derivations and theory.

**Review Assessment: Checking Correctness Of Experiments:**

I assessed the sensibility of the experiments.

**Review Assessment: Thoroughness In Paper Reading:**

I made a quick assessment of this paper.

---

### Decision · Program_Chairs · 2019-12-19

**Decision:**

Reject

**Comment:**

The paper focuses on extracting the underlying dynamics of objects in video frames, for background/foreground extraction and video classification. Generally speaking, the presentation of the paper should be improved. Novelty should be clarified, contrasting the proposed approach with existing literature. All reviewers also agree the experimental section is also too weak in its current form.